# The Co-Application of PGPR and Biochar Enhances the Production Capacity of Continuous Cropping Peppers in the Karst Yellow Soil Region of Southwest China

Meng Zhang [1], Yanling Liu [1], Quanquan Wei [1], Jiulan Gou [1,*], Lingling Liu [1], Xiaofeng Gu [1] and Ming Wang [2]

1 Institute of Soil and Fertilizer, Guizhou Academy of Agricultural Sciences, Guiyang 550006, China
2 College of Eco-Environmental Engineering, Guizhou Minzu University, Guiyang 550006, China
* Correspondence: gzgoujiulan@163.com

**Abstract:** In recent years, a significant impediment to the advancement of China's agricultural sector is the noteworthy challenge posed by diminished crop yields and quality due to ongoing continuous cropping obstacles. Numerous studies have consistently showcased the potential of plant growth-promoting rhizobacteria (PGPR) and biochar in augmenting the alleviation of continuous cropping barriers. Nevertheless, the potential of PGPR and biochar to remediate and improve continuous cropping peppers in the karst yellow soil area remains unclear. A 2-year field experiment was implemented to examine the impact of PGPR and biochar, when applied alone or in combination, on the production potential of continuous cropping peppers. The results revealed that PGPR and biochar significantly elevated the yield of fresh and dry pepper compared with TF treatment. The utilization of PGPR and biochar resulted in an augmentation of free amino acids, soluble sugar, and vitamin C content in pepper fruits, but a reduction in the nitrate content, which proved advantageous in enhancing the overall quality of peppers. Furthermore, the use of PGPR and biochar demonstrated significant benefits in enhancing NPK accumulation, fertilizer utilization, and economic efficiency. Nevertheless, the co-application of PGPR and biochar yielded significantly better results compared to their individual application. In conclusion, the utilization of PGPR and biochar demonstrated a favorable impact on the productivity and economic benefits of continuous cropping peppers. The simultaneous application of PGPR and biochar represents a promising approach to enhancing yield and improving the quality of peppers in the karst yellow soil region of Southwest China.

**Keywords:** plant growth-promoting rhizobacteria; biochar; yield and quality; fertilizer efficiency; economic benefits





## 1. Introduction

Modern agriculture is confronting novel challenges, and the escalating demand for agricultural goods strains crop production and compels agricultural practices to mitigate their adverse ecological effects [1,2]. Thus, enhancing the sustainable productivity of existing farmlands becomes pivotal to fulfilling forthcoming global crop needs while minimizing environmental detriments. The intensified cultivation of monoculture cash crops has led to complications involving continuous cropping and the emergence of replanting-related diseases, profoundly impacting the viability of cultivable terrain. This situation underscores a substantial peril to both regional food security and environmental integrity [3–5]. Numerous researchers have substantiated that the dysregulation of soil microbial diversity stands as the central factor driving challenges in continuous cropping obstacles [6–9]. In recent years, researchers have increasingly embraced the utilization of beneficial microorganisms and organic amendments to rehabilitate compromised soil in agricultural contexts, and this approach aims to enhance crop yield and quality while improving the soil microenvironment [10–12].

Plant growth-promoting rhizobacteria (PGPR) are beneficial microorganisms that reside in the soil microflora of the plant rhizosphere. Specifically, PGPR possess numerous functions, including biological nitrogen fixation, phosphorus and potassium solubilization, antibiotic secretion, and hormone synthesis [13,14]. It has been demonstrated that PGPR can colonize plant roots and enhance the solubilization of insoluble nutrients by releasing organic acids. This colonization and nutrient solubilization contribute to promoting plant growth and improving the host's uptake and utilization of mineral nutrients [15]. Notably, studies have revealed that PGPR can produce antibiotics, antimicrobial proteins, pathogen cell wall hydrolases, and other biologically active components throughout their growth and reproduction, which can effectively inhibit or eradicate plant pathogenic bacteria [16,17]. Moreover, PGPR has been shown to enhance plant root growth and development by producing growth hormones and cytokinins. It also improves nutrient uptake and helps maintain balanced nutrition in crops [18,19]. Consequently, the utilization of PGPR has been widely recognized as the most effective approach for mitigating the challenges associated with continuous cropping and boosting crop productivity. Similar studies have been conducted using biochar. Biochar can improve soil fertility and reduce the occurrence of diseases and pests by improving the soil microbial environment [20–25]. At present, PGPR and biochar are extensively utilized as vital soil amendment materials in agricultural production [26,27].

China stands as a global leader in pepper production. Nonetheless, the challenge of continuous cropping hurdles has pervaded pepper cultivation regions due to the adoption of unsustainable farming and management methods. The long-term continuous cultivation of pepper has fostered an escalation in pest and disease occurrences. As a consequence, both pepper yield and quality have experienced a consistent decline, exerting an adverse effect on economic returns. This predicament significantly hampers the sound and sustainable progression of pepper agriculture [28]. While numerous experiments have underscored the positive influence of individual PGPR or biochar applications, boosting crop yield and enhancing soil quality, it is worth noting that the majority of these experiments were carried out in controlled laboratory or greenhouse settings. Consequently, research encompassing field experiments remains relatively scarce. In this research, we hypothesized that the application of PGPR and biochar would have an ameliorative effect on continues cropping peppers, so a 2-year field experiment was implemented to investigate the potential ameliorative effect of applying PGPR and biochar either alone or in combination on pepper fields that have been continuously planted for 5 years, with the following aims: (1) to observe the effects of PGPR and biochar on fresh yield, dry yield, and the quality of the peppers, (2) to examine the impacts of PGPR and biochar on NPK accumulation and fertilizer use efficiency, and (3) to calculate the effects of applying PGPR and biochar on improving the economic output of peppers.

## 2. Materials and Methods

### 2.1. Site Description

The field experiment was implemented from 2021 to 2022 in Guizhou Province of China. Prior to the experiment, the field was cultivated with peppers for a period of five years, spanning 2016–2020. The experimental region is characterized by the presence of yellow soil, which is extensively found in the karst mountains of Southwest China. The yellow soil type at this region was classified as Acrisol in the World Reference Base for Soil Resources (WRB), and was developed from the Triassic limestone and sand shale efflorescence. Table 1 presents the fundamental physicochemical properties of the soil in the experimental region.

**Table 1.** The basic physicochemical properties of soil and biochar.

|  | pH | Organic Carbon (g·kg$^{-1}$) | Total Nitrogen (g·kg$^{-1}$) | Total Phosphorus (g·kg$^{-1}$) | Available Phosphorus (mg·kg$^{-1}$) | Total Potassium (g·kg$^{-1}$) | Available Potassium (mg·kg$^{-1}$) |
|---|---|---|---|---|---|---|---|
| Soil | 6.19 | 11.88 | 1.36 | 1.02 | 17.11 | 14.72 | 156.83 |
| Biochar | 8.23 | 354.86 | 39.84 | 9.23 | 275.47 | 19.88 | 748.55 |

### 2.2. Experimental Materials

The experimental pepper variety was 'Zunla 9', a locally cultivated pod pepper variety. The chemical fertilizer used in the experiment is the best-selling compound fertilizer in the local area, with a total nutrient content of 42% (N content of 18%, $P_2O_5$ content of 6%, $K_2O$ content of 18%). PGPR solution was from Shandong Lvlong Biotechnology Co., Ltd. (Zhucheng, China), and biochar was from Guizhou Jinyefeng Ecological Agricultural Technology Co., Ltd. (Weining, China). The PGPR solid was produced using a hybrid strain comprising *Bacillus amyloliquefaciens* and *Paenibacillus polymyxa*. The effective viable bacterial count exceeded $15 \times 10^8$ CFU·g$^{-1}$. The raw material for producing biochar was distillers' grains processed at 550 °C. Table 1 lists the fundamental physicochemical properties of biochar.

### 2.3. Experimental Design

Two pepper planting seasons were included in the field experiments conducted from 2021 to 2022. The study included six treatments, each with three replicates. The detailed treatments were shown in Table 2. It should be noted that the application rate of chemical fertilizers was provided by the local agricultural department. In the TFP, TFB, and TFPB1 treatments, the application rates of PGPR (15.00 kg·hm$^{-2}$) and biochar (1500.00 kg·hm$^{-2}$) were also determined based on the recommended application rates of organic remediation materials by the local agricultural department. In the TFPB2 treatment, PGPR and biochar were applied twice as much as in the TFPB treatment, with the aim of exploring whether increasing the application rate of both has a greater potential for improvement when farmers' economic conditions allow.

**Table 2.** The application rates of chemical fertilizer, PGPR, and biochar in different treatments.

| Treatments | Chemical Fertilizer (kg·hm$^{-2}$) | PGPR (kg·hm$^{-2}$) | Biochar (kg·hm$^{-2}$) |
|---|---|---|---|
| CK | 0 | 0 | 0 |
| TF | 1500.00 | 0 | 0 |
| TFP | 1500.00 | 15.00 | 0 |
| TFB | 1500.00 | 0 | 1500.00 |
| TFPB1 | 1500.00 | 15.00 | 1500.00 |
| TFPB2 | 1500.00 | 30.00 | 3000.00 |

In the experiment, prior to transplanting pepper plants, chemical fertilizer and biochar were utilized in the soil as a basal fertilizer. Afterward, a rotary tiller was utilized to blend the fertilizers with the soil thoroughly. Fifteen days after ridging, pepper seedlings were transplanted and then covered with a plastic film. PGPR was applied 20 days after transplanting the pepper seedlings. For the TFP and TFPB1 treatments, PGPR was applied by dissolving 1 kg of PGPR in 300 L of water. The solution was then allowed to stand for 6 h before being poured onto the roots of the peppers. The watering volume was 100 mL·plant$^{-1}$. For the TFPB2 treatment, 2 kg of PGPR was dissolved in 300 L of water, keeping the other operational steps consistent with the abovementioned method. Furthermore, the PGPR solution was applied through irrigation in the TFP, TFPB1, and TFPB2 treatments. Similarly, the roots of peppers in the CK, TF, and TFB treatments were irrigated with water using the same method, with a watering volume of 100 mL plant$^{-1}$. This method was employed to



mitigate variations in pepper growth stemming from disparities in water supply. Referring to the local recommended planting density for pod peppers, it was $4.5 \times 10^4$ plants·ha$^{-1}$. Each treatment was randomly assigned to a block for the experiment, and carried out in a plot area measuring 40.50 m$^2$. Moreover, consistent field managements were employed to guarantee the precision of the test outcomes.

### 2.4. Soil Sampling and Analysis

Prior to the application of fertilizers, soil samples were collected from 15 randomly selected sites utilizing a soil auger, at a depth of 0–20 cm. We prepared soil samples according to the method of Ning et al. [29], and determined the physicochemical properties of the soil.

### 2.5. Plant Sampling and Analysis

In the pod pepper's maturation phases, six plants were gathered from each plot. We prepared plant samples according to the method of Zhang et al. [28], followed by digestion using a mixture of concentrated H$_2$SO$_4$ and H$_2$O$_2$ to analyze the concentrations of N, P, and K [28]. Additionally, the fresh samples were harvested from each plot to analyze the levels of reducing sugars, VC, free amino acids, and nitrates [28].

### 2.6. Pepper Yield

The quantity of newly harvested pod peppers in each plot was evaluated according to the ripeness of the pods. The ultimate yield of fresh peppers was ascertained by gauging the combined weight across multiple harvests. Additionally, the moisture content of freshly collected pod peppers was established by subjecting them to laboratory drying. Following this, the yield of dehydrated pod peppers was computed.

### 2.7. Calculations

We calculated the NPK nutrient accumulation, fertilizer utilization efficiency, and economic benefits using the previous method [28]. It should be noted that the price per kilogram of dried pod pepper was CNY 20.00 in assessing economic benefits, and the cost of chemical fertilizer and biochar was 3350 and 2000 CNY·t$^{-1}$, respectively. The PGPR was 20.00 CNY·kg$^{-1}$.

### 2.8. Statistical Analysis

Single- and multiple-factor analysis of variance (ANOVA) were performed using the SPSS 20.0 software (SPSS Inc., Chicago, IL, USA). Duncan's method was utilized for multiple comparisons at $p < 0.05$. The figures were created using Origin 8.0 software (OriginLab Corporation, Northampton, MA, USA).

## 3. Results

### 3.1. The Impact of Applying PGPR and Biochar on the Yield

The application of PGPR and biochar positively affected the pepper yield, as illustrated in Figure 1. When compared to TF treatment, the utilization of PGPR and biochar, either separately or in combination (TFP, TFB, TFPB1, and TFPB2), resulted in an increase in the yield of fresh peppers by 11.52–49.68% (2021) and 20.99–66.86% (2022). The TFPB2 treatment yielded the highest amount of fresh peppers over the span of two years, particularly in 2022, where it reached a maximum of 16,315 kg·ha$^{-1}$. Furthermore, the findings indicated a similarity in the changes observed in both dry pepper yield and fresh pepper yield. In comparison to TF treatment, the applying PGPR and biochar alone or in combination resulted in a significant increase in dry pepper yield by 2.09–53.08% (2021) and 34.43–75.44% (2022), respectively. In both 2021 and 2022, the TFPB2 treatment demonstrated the highest yield of dry peppers.

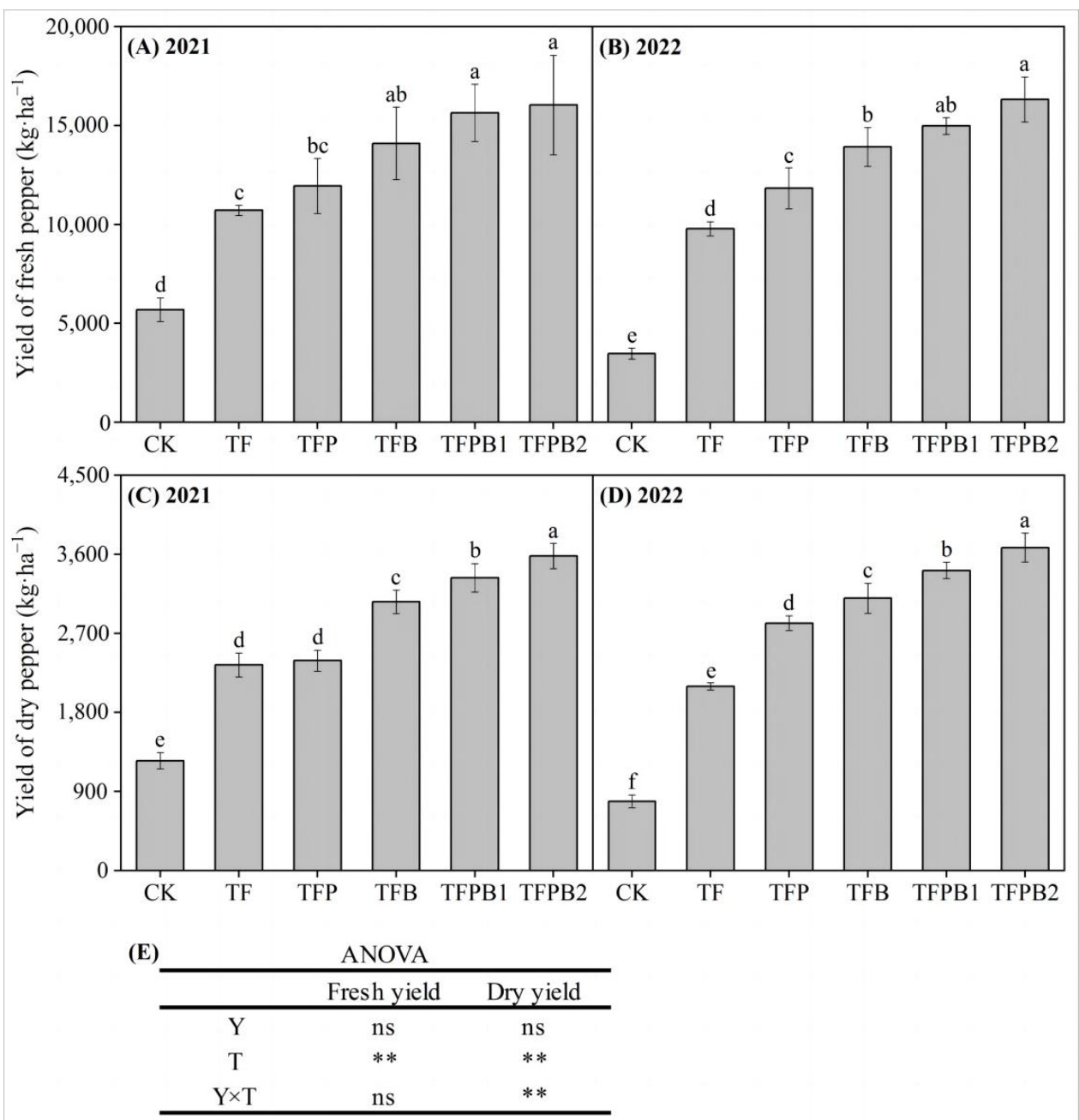

**Figure 1.** The yield of fresh (**A,B**) and dry (**C,D**) peppers, and ANOVA (**E**) in different treatments. In (**A–D**), the different lowercase letters indicate significant differences among different treatments at $p < 0.05$ in the same year. In (**E**), Y represents year, T represents different treatments, and Y×T represents the interaction between year and treatment. ns represents no difference, ** represents statistical significance at $p < 0.01$. The same applies below.

### 3.2. The Impact of Applying PGPR and Biochar on the Quality of Fresh Pepper Fruits

The quality of fresh pepper fruits was improved by applying PGPR and biochar (Table 3). The results exhibited no significant difference in the content of free amino acids among all treatments in 2021. However, in 2022, the TFPB2 treatment exhibited a significantly higher free amino acid content than the TF treatment. In comparison to TF treatment, the applying PGPR and biochar alone or in combination led to a significant increase in reducing sugar content, with an increase of 24.90–50.87% in 2021 and 37.63–70.82% in 2022. The application of PGPR and biochar in combination (TFPB1 and TFPB2) resulted in an increase of 7.56–20.79% in reducing sugar content in 2021 and 7.42–24.12% in 2022, as

compared to the PGPR and biochar treatment alone (TFP and TFB). In the meantime, the utilization of PGPR and biochar alone or in combination resulted in an increase in VC content of 8.42–32.63% in 2021 and 30.59–56.47% in 2022. Furthermore, the application of either PGPR or biochar alone, or their combination, resulted in a notable decrease in the nitrate content of fresh fruits by 16.24–21.61% in 2021 and 21.02–41.56% in 2022, compared to TF treatment. Notably, the combination of PGPR and biochar exhibited the most significant impact.

**Table 3.** The fruit quality of fresh pepper in different treatments.

| Year | Treatments | Free Amino Acid (g·kg$^{-1}$) | Reducing Sugar (mg·kg$^{-1}$) | VC (g·kg$^{-1}$) | Nitrate (mg·kg$^{-1}$) |
|------|-----------|------------------------------|-------------------------------|------------------|------------------------|
| 2021 | CK    | 3.70 ± 0.10 a  | 29.06 ± 0.91 f | 0.69 ± 0.04 e | 78.98 ± 1.64 b  |
|      | TF    | 3.74 ± 0.10 a  | 32.73 ± 1.16 e | 0.95 ± 0.05 d | 84.19 ± 2.28 a  |
|      | TFP   | 3.77 ± 0.09 a  | 40.88 ± 1.65 d | 1.03 ± 0.04 c | 70.52 ± 2.70 c  |
|      | TFB   | 3.79 ± 0.12 a  | 43.54 ± 0.78 c | 1.11 ± 0.04 b | 67.95 ± 2.61 cd |
|      | TFPB1 | 3.80 ± 0.09 a  | 46.83 ± 1.30 b | 1.21 ± 0.03 a | 66.35 ± 2.62 cd |
|      | TFPB2 | 3.83 ± 0.12 a  | 49.38 ± 1.12 a | 1.26 ± 0.04 a | 66.00 ± 2.97 d  |
| 2022 | CK    | 3.56 ± 0.07 c  | 24.08 ± 1.01 e | 0.63 ± 0.03 e | 76.81 ± 3.14 b  |
|      | TF    | 3.69 ± 0.04 bc | 30.40 ± 0.80 d | 0.85 ± 0.03 d | 84.43 ± 2.81 a  |
|      | TFP   | 3.79 ± 0.12 ab | 41.84 ± 1.21 c | 1.11 ± 0.02 c | 66.68 ± 3.35 c  |
|      | TFB   | 3.80 ± 0.09 ab | 44.58 ± 0.66 c | 1.14 ± 0.04 c | 64.49 ± 3.28 cd |
|      | TFPB1 | 3.81 ± 0.14 ab | 47.89 ± 2.48 b | 1.23 ± 0.04 b | 58.82 ± 3.40 d  |
|      | TFPB2 | 3.96 ± 0.09 a  | 51.93 ± 2.59 a | 1.33 ± 0.03 a | 49.34 ± 4.07 e  |
| ANOVA | | | | | |
| Y    |  | ns | ns | ns | ** |
| T    |  | ** | ** | ** | ** |
| Y×T  |  | ns | ** | ** | ** |

Note: The different lowercase letters indicate significant differences among different treatments at *p* < 0.05 in the same year. ns represents no difference, ** represents statistical significance at *p* < 0.01.

### 3.3. The Impact of Applying PGPR and Biochar on the NPK Accumulation

The application of PGPR and biochar exhibited a positive impact on the accumulation of NPK nutrients (Figure 2). Relative to the TF treatment, the application of PGPR and biochar, either individually or combined, resulted in an increase of 14.10–70.88%, 4.13–55.08%, and 12.20–49.23% in N, P, and K accumulation in 2021. Moreover, in 2022, the increase reached 42.82–125.60%, 40.54–143.95%, and 41.30–96.94% for the same nutrients. In 2021, the co-application of PGPR and biochar treatments (TFPB1 and TFPB2) resulted in an increase of N, P, and K accumulation by 13.23–49.77%, 16.66–48.93%, and 11.21–33.01%, respectively, compared to application alone (TFB and TFV). Similarly, in 2022, these combined treatments showed an increase of 13.70–57.95%, 15.12–73.58%, and 13.55–39.38% for N, P, and K accumulation, respectively.

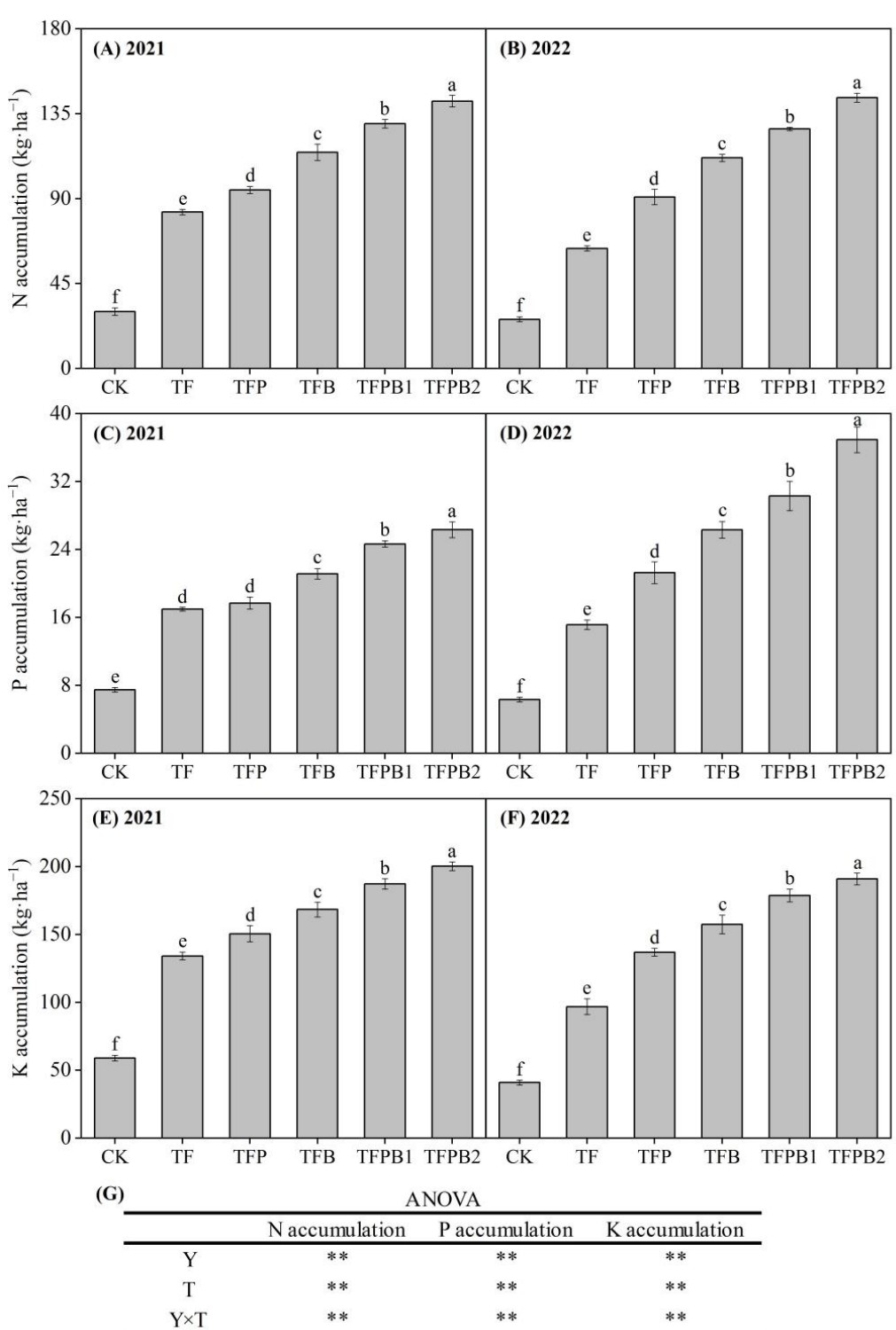

**Figure 2.** The N accumulation (**A**,**B**), P accumulation (**C**,**D**), K accumulation (**E**,**F**) in different treatments, and ANOVA (**G**). In (**A**–**F**), the different lowercase letters indicate significant differences among different treatments at $p < 0.05$ in the same year. In (**G**), ** represents statistical significance at $p < 0.01$.

*3.4. The Impact of Applying PGPR and Biochar on Agronomic and Recovery Efficiency of NPK*

Table 4 demonstrates that the utilization of PGPR and biochar resulted in a significant enhancement in fertilizer efficiency. Compared to the TF treatment, the application of PGPR and biochar alone or in combination resulted in an increase of $AE_N$, $AE_P$, and $AE_K$ by 4.46–113.86%, 4.46–113.86%, and 4.46–113.86% in 2021. In 2022, the increase was even higher, ranging from 55.05–120.62%, 55.09–120.77%, and 55.05–120.62%, respectively. In 2021, the $RE_N$, $RE_P$, and $RE_K$ values following PGPR and biochar application either alone or in combination were 1.22–2.11 times, 1.07–1.98 times, and 1.22–1.88 times higher than those of the TF treatment in 2021. In 2022, these values were 1.72–3.12 times, 1.70–3.48 times, and 1.71–2.68 times higher than those of the TF treatment, respectively. The co-application of PGPR and biochar treatments (TFPB1 and TFPB2) resulted in an average increase of 7.55% in REN, 8.83% in REP, and 13.32% in REK when compared to their individual application (TFP and TFB).

**Table 4.** The fertilizer utilization of different treatments.

| Year | Treatments | AE (kg·kg$^{-1}$) | | | RE (%) | | |
|---|---|---|---|---|---|---|---|
| | | **AE$_N$** | **AE$_P$** | **AE$_K$** | **RE$_N$** | **RE$_P$** | **RE$_K$** |
| 2021 | CK | — | — | — | — | — | — |
| | TF | 4.04 ± 0.33 c | 12.12 ± 1.00 c | 4.04 ± 0.33 c | 19.54 ± 1.09 e | 10.56 ± 0.55 d | 27.90 ± 1.82 e |
| | TFP | 4.22 ± 0.27 c | 12.66 ± 0.82 c | 4.22 ± 0.27 c | 23.87 ± 1.23 d | 11.34 ± 0.60 d | 33.96 ± 2.49 d |
| | TFB | 6.70 ± 0.80 b | 20.11 ± 2.41 b | 6.70 ± 0.80 b | 31.27 ± 2.04 c | 15.17 ± 0.77 c | 40.55 ± 2.76 c |
| | TFPB1 | 7.72 ± 0.62 a | 23.15 ± 1.85 a | 7.72 ± 0.62 a | 36.89 ± 1.47 b | 19.08 ± 0.28 b | 47.54 ± 1.90 b |
| | TFPB2 | 8.64 ± 0.38 a | 25.92 ± 1.13 a | 8.64 ± 0.38 a | 41.30 ± 0.55 a | 20.95 ± 1.07 a | 52.37 ± 0.42 a |
| 2022 | CK | — | — | — | — | — | — |
| | TF | 4.85 ± 0.28 e | 14.54 ± 0.85 e | 4.85 ± 0.28 e | 13.93 ± 0.82 e | 9.78 ± 0.94 e | 20.77 ± 1.74 e |
| | TFP | 7.52 ± 0.57 d | 22.55 ± 1.70 d | 7.52 ± 0.57 d | 24.01 ± 1.71 d | 16.60 ± 1.76 d | 35.59 ± 1.07 d |
| | TFB | 8.57 ± 0.64 c | 25.70 ± 1.93 c | 8.57 ± 0.64 c | 31.71 ± 1.01 c | 22.21 ± 1.37 c | 43.16 ± 1.84 c |
| | TFPB1 | 9.74 ± 0.18 b | 29.21 ± 0.55 b | 9.74 ± 0.18 b | 37.37 ± 0.82 b | 26.63 ± 2.00 b | 51.06 ± 1.63 b |
| | TFPB2 | 10.70 ± 0.53 a | 32.10 ± 1.58 a | 10.70 ± 0.53 a | 43.51 ± 1.30 a | 33.99 ± 1.50 a | 55.57 ± 0.83 a |
| ANOVA | | | | | | | |
| Y | | ** | ** | ** | ns | ** | ns |
| T | | ** | ** | ** | ** | ** | ** |
| Y×T | | ** | ** | ** | ** | ** | ** |

Note: $AE_N$ stands for agronomic efficiency of N, $AE_P$ stands for agronomic efficiency of P, $AE_K$ stands for agronomic efficiency of K. $RE_N$ stands for recovery efficiency of N, $RE_P$ stands for recovery efficiency of P, $RE_K$ stands for recovery efficiency of K. The different lowercase letters indicate significant differences among different treatments at $p < 0.05$ in the same year. ns represents no difference, ** represents statistical significance at $p < 0.01$.

*3.5. The Impact of Applying PGPR and Biochar on Output Value and Net Income*

In comparison with TF treatment (Table 5), the application of PGPR and biochar, either alone or in combination, resulted in an increase in the OV of dry peppers by 2.09–53.08% (2021) and 34.43–75.44% (2022), respectively. Furthermore, compared to TF treatment, the net income of PGPR and biochar, either alone or in combination, increased by 1.63–43.90% in 2021 and 38.55–68.24% in 2022. The co-application of PGPR and biochar (TFPB1 and TFPB2) resulted in a 9.77–41.58% increase in net income in 2021 and an 11.20–21.43% increase in 2022, as compared to the application of PGPR or biochar alone (TFP and TFB). Among all treatments, the TFPB2 treatment exhibited the highest net income for dry peppers over the 2-year period, reaching 59,788 CNY·ha$^{-1}$ in 2021 and 61,664 CNY·ha$^{-1}$ in 2022.

**Table 5.** The economic benefits of different treatments.

| Year | Treatments | OV (CNY·ha$^{-1}$) | FV (CNY·ha$^{-1}$) | NEI (CNY·ha$^{-1}$) |
|------|-----------|-----|-----|-----|
| 2021 | CK | 24,990 ± 1829 e | — | 24,990 ± 1829 d |
| | TF | 46,799 ± 2778 d | 5250 | 41,549 ± 2778 c |
| | TFP | 47,778 ± 2409 d | 5550 | 42,228 ± 2409 c |
| | TFB | 61,185 ± 2686 c | 8250 | 52,935 ± 2686 b |
| | TFPB1 | 66,659 ± 3238 b | 8550 | 58,109 ± 3238 a |
| | TFPB2 | 71,638 ± 2897 a | 11850 | 59,788 ± 2897 a |
| 2022 | CK | 15,737 ± 1433 f | — | 15,737 ± 1433 d |
| | TF | 41,903 ± 815 e | 5250 | 36,653 ± 815 c |
| | TFP | 56,332 ± 1663 d | 5550 | 50,782 ± 1663 b |
| | TFB | 62,004 ± 3430 c | 8250 | 53,754 ± 3430 b |
| | TFPB1 | 68,324 ± 1845 b | 8550 | 59,774 ± 1845 a |
| | TFPB2 | 73,514 ± 3301 a | 11850 | 61,664 ± 3301 a |
| ANOVA | | | | |
| Y | | ns | — | ns |
| T | | ** | — | ** |
| Y×T | | ** | — | ** |

Note: OV means output value, FV means fertilizer value, NEI means net income. The different lowercase letters indicate significant differences among different treatments at $p < 0.05$ in the same year. ns represents no difference, ** represents statistical significance at $p < 0.01$. — represents non-exist.

## 4. Discussion

The study findings demonstrated that PGPR and biochar application, individually or in combination, positively impacted the yield of continuous cropping peppers. Relative to the TF treatment, the yield of fresh peppers increased by 11.52–49.68% in 2021 and 20.99–66.86% in 2022. Meanwhile, the yield of dry peppers increased by 2.09–53.08% in 2021 and 34.43–75.44% in 2022. Studies have shown that PGPR has a significant impact on improving the microbial community structure of rhizosphere soil and enhancing the functional diversity of microorganisms, and such function promotes the growth of beneficial bacteria while inhibiting the growth of pathogenic bacteria [16,30]. Simultaneously, PGPR can create an optimal micro-ecological environment for the plant root system to augment water and fertilizer absorption capacity, thereby facilitating plant growth [31]. Moreover, the beneficial impacts of biochar on crop yields can be ascribed to various factors. Studies have demonstrated that the application of biochar can enhance soil quality. Specifically, biochar can not only addresses the issue of soil compaction caused by excessive fertilizer usage, but also promotes higher crop yields [32–36]. Significantly, utilizing biochar can heighten soil microorganism activity and enhance the microbiological setting, leading to improved conditions for crop root development and yield [37–39].

Interestingly, the co-application of PGPR and biochar in this study brought about more significant increases in yield compared to individual applications alone, demonstrating a synergistic effect of the co-application of PGPR and biochar. Previous studies also supported our conclusion. These studies highlighted that the combination of PGPR and biochar could be effective in enhancing the nutrient status and microbial environment of continuous crop barrier soils. The inhibition of harmful pathogens was identified as a significant mechanism [40,41]. This phenomenon can be attributed to the dual stimulating effects of PGPR and biochar [42,43]. On the one hand, PGPR can fulfill important functions, such as solubilizing phosphorus and potassium, fixing nitrogen, and decomposing effective nutrients in the soil. Additionally, they facilitate the release and sequestration of trace elements, thereby enhancing soil nutrient cycling [44,45]. On the other hand, the porous structure of biochar creates an ideal growth environment for PGPR, facilitating their propagation in the rhizosphere soil. This, in turn, leads to significant alterations in soil microbial function and community diversity [46,47].

The quality of fruit serves as a crucial criterion for evaluating the overall quality of agricultural products. The results showed the application of PGPR and biochar increased the levels of fresh pepper quality (Table 3). These results uncovered that the implementation of effective improvement measures positively influenced the quality enhancement of continuous cropping peppers. It has been found that PGPR can induce plant resistance by producing a variety of phytohormones, such as organic acids, gibberellins, and cytokinins [48]. They not only stimulate the accumulation of soluble proteins, proline, and other substances to counteract external environmental stress, but also enhance crop quality [49,50]. Moreover, the use of biochar encourages the harmonization and equilibrium of nutrient metabolism within crops due to its stable fertilization and continuous nutrient release, resulting in improved fruit quality [51,52]. Additionally, the enhancement in fruit quality may be attributed to the utilization of PGPR or biochar, which has been shown to increase the photosynthetic rate of leaves and facilitate the translocation of photosynthetic products to the fruit [27,53]. This process has also been found to have a positive impact on fruit quality. Notably, this study also observed the synergistic benefits of PGPR and biochar in enhancing the quality of continuous cropping peppers, which could be attributed to their complementary effects [47,54]; however this requires further research.

In this research, the application of PGPR and biochar, either alone or in combination, led to a significant increase in the accumulation of NPK (Figure 2) and a significant improvement in fertilizer utilization efficiency (Table 4). Research has shown that PGPR can expedite the decomposition of organic matter in the soil, facilitate the dissolution of insoluble nutrients, and subsequently release available nutrients to enhance the efficacy of soil nutrients. This process is also beneficial for the absorption and accumulation of mineral nutrients in plants [55,56]. In addition, research has indicated that the utilization of biochar enhances the soil carbon-to-nitrogen ratio and suppresses nitrogen conversion and denitrification through soil microorganisms, thereby facilitating the retention of $NH_4^+$ and $NO_3^-$ in the soil [57,58]. Moreover, biochar can serve as a substitute for conventional phosphorus fertilizers due to its naturally high phosphorus content, which can also modify the dynamics and effectiveness of phosphorus in soils through processes like phosphorus adsorption and desorption, as well as by regulating the composition of the soil microbial community [59–61]. Furthermore, aside from directly contributing to potassium levels, biochar has the potential to enhance soil potassium content and boost the efficiency of potassium fertilizer utilization through the promotion of microbial activity [62,63]. Notably, the co-application of PGPR and biochar results in a further enhancement of nutrient accumulation and utilization by the plants, as compared to their individual application. This can be attributed to the enhanced colonization ability and survival rate of PGPR in rhizosphere soil through the addition of biochar [64,65]. Furthermore, the synergistic interaction between biochar and PGPR amplifies the capability of functional microorganisms to regulate nutrient transformations and cycling in the rhizosphere soil. This, in turn, promotes the accumulation of nutrients by plants and improves nutrient utilization efficiency [54,66]. We hypothesize that the synergistic mechanism of PGPR and biochar is mainly realized through the improvement of the soil microbial environment (including soil microorganisms and soil enzyme activities, etc.), and although the soil was not evaluated in this study, this will be the focus of our later research.

Importantly, it should be highlighted that the TFP treatment exhibited comparatively suboptimal results concerning both biological and economic aspects in the context of continuous pepper cropping, implying that relying solely on the application of PGPR was not an ideal approach. This could be attributed to the adherence of PGPR to the soil particles' surface, which hindered the target microorganisms from colonizing the deep soil layers [67,68]. Consequently, this significantly impaired their ability to colonize the root surface [16,69]. Furthermore, the colonization of PGPR is influenced by various soil environmental factors, including temperature, oxygen, and moisture, as well as competition from native microorganisms [54,70]. Microbial immobilization technology is one approach to addressing the aforementioned issues, with biochar being recognized as an exceptional

carrier for microorganism immobilization [71,72]. Hence, future studies should prioritize strengthening research on biochar-immobilized PGPR composites to enhance the soil remediation function of biochar and PGPR.

### 5. Conclusions

The study findings demonstrated that applying PGPR and biochar, either alone or in combination, resulted in increased productivity, improved fruit quality, enhanced fertilizer utilization, and enhanced economic benefits of continuous cropping peppers in the karst region of southwest China. In the current conditions, the synergistic use of PGPR and biochar emerged as the most effective strategy for boosting productivity and economic advantages in the continuous cropping peppers. Thus, the synergistic use of PGPR and biochar (TFPB1) is recommended to alleviate the issues of continuous cropping peppers grown in the karst yellow soil region of Southwest China. In addition, higher PGPR and biochar (TFPB2) can be applied to increase the production potential of continuous chili peppers under better economic conditions.

**Author Contributions:** M.Z. and J.G. designed the experiments and conducted the research; Y.L., Q.W., X.G., L.L. and M.W. performed most of the experiments and analyzed the data; M.Z. wrote the article. J.G. and M.W. provided recommendations for experiment conduct. All authors have read and agreed to the published version of the manuscript.

**Funding:** This work was funded by the Guizhou Provincial Basic Research Program (Natural Science) (QKHJC-ZK[2023]ZD022) and the Science and Technology Innovation Special Project of Guizhou Academy of Agricultural Sciences (No.[2023]13).

**Data Availability Statement:** The original contributions presented in the study are included in the article. Further inquiries can be directed to the corresponding author.

**Conflicts of Interest:** The authors declare that the research was conducted in the absence of any commercial or financial relationships that could be construed as a potential conflicts of interest.

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
