# Peer review of "The Co-Application of PGPR and Biochar Enhances the Production Capacity of Continuous Cropping Peppers in the Karst Yellow Soil Region of Southwest China"

_horticulturae, doi:10.3390/horticulturae9101104_

Round 1

Reviewer 1 Report

The manuscript presents a research evidence of the improvement of the soil and crop production after amendment with PGPR or/and biochart.

I haven't detected any important fail in the manuscript, except a few typing mistakes that need to be corrected. I include the PDF manuscript with the comments and corrections needed.

Nevertheless, the authors have to include the proper statistical analysis to support the year-on-year differences found in the TF treatments assuring that continuous cropping results in a decrease of the yield.

Author Response

Thank you very much for taking your time to review this manuscript and give us another chance to revise the manuscript. We will cherish the opportunity of this revision. ​Your comments make this paper completer and more outstanding! I really appreciate all your comments and suggestions! Please find my itemized responses in below and my revisions/corrections in the re-submitted files. Thanks again.

Q1: It would be very good to include here briefly the hypothesis of the work.

A1: Thank you very much for your suggestion. We have added the hypothesis of the work in the manuscript.

Q2: Vermicompost is not included in the table and this is the first time this is mentioned. Please, clarify it.

A2: Thank you very much for your suggestion. We have corrected the error.

Q3: Is this equivalent to pod pepper production? Nutrient accumulation could be misinterpreted.

A3: Thank you very much for your suggestion. We have modified the words.

Q4: Although this can be inferred from the data included in Figure 1, I can't see if these differences are or not statistically significant. To support this, the authors should include an interannual statistical analysis of the yield values.

A4: Thank you very much for your suggestion. We have added the interaction analysis between years and treatments of all data.

Q5: Single f?

A5: Thank you very much for your suggestion. We have corrected the error.

Reviewer 2 Report

Paper “Co-application of PGPR and biochar enhances the production capacity of continuous cropping peppers in the karst yellow 3 soil region of Southwest China” investigates effects of PGPR and biochar application on the yield and quality of pepper. The investigation is conducted through long-term field experiment, which makes it rare and useful insight into effects of PGPR and biochar application in real conditions. However, although generally well-written, paper is a bit over-lenght due to certain repetitiveness of general findings, and this should be adequately addressed through text. Additionally, certain methodological issues are rising as well as synthesis in text, which could be improved. I suggest major revision of the paper prior to acceptance in Horticulture journal.

Line 18-19: Why there are no single treatments with TF combined with PGPR at a rate of 30 kg ha-1 and TF combined with biochar at a rate of 3000 kg ha-1? Why only combined effects of this treatment were accessed, contrary to the effects of PGPR at a rate of 15 kg ha-1 and TF combined with biochar at a rate of 1500 kg ha-1?That way reaction concerning gradient of PGPR and biochar content could be examined. Especially since the highest applied rate was recommended for use in the Conclusion section.

Line 97: Aims and hypothesis of this research could be clearly stated at the end of Introduction section.

Line 107: can you also use additional classification of “yellow soil”, e.g. according to World Referenc Base for Soil Resources (WRB)?

Discussion section should be shortened and overwritten, with only relative facts kept. Much of the text relates to the general conclusions, without specific direct effects in particular experimental case. More comparisons with similar research should be made in this section. No direct synergistic mechanism of biochar and PGPR in this research was revealed. More clarification and direct connection of soil amendments and measured effects should be provided.

Author Response

Thank you very much for taking your time to review this manuscript and give us another chance to revise the manuscript. We will cherish the opportunity of this revision. ​Your comments make this paper completer and more outstanding! I really appreciate all your comments and suggestions! Please find my itemized responses in below and my revisions/corrections in the re-submitted files. Thanks again.

Q1: Line 18-19: Why there are no single treatments with TF combined with PGPR at a rate of 30 kg ha-1 and TF combined with biochar at a rate of 3000 kg ha-1? Why only combined effects of this treatment were accessed, contrary to the effects of PGPR at a rate of 15 kg ha-1 and TF combined with biochar at a rate of 1500 kg ha-1?That way reaction concerning gradient of PGPR and biochar content could be examined. Especially since the highest applied rate was recommended for use in the Conclusion section.

A1: Thank you very much for your suggestion. We have added an explanation for this issue in the experimental design.

Q2: Line 97: Aims and hypothesis of this research could be clearly stated at the end of Introduction section.

A2: Thank you very much for your suggestion. We have added the hypothesis and aims of this research in the manuscript.

Q3: Line 107: can you also use additional classification of “yellow soil”, e.g. according to World Referenc Base for Soil Resources (WRB)?

A3: Thank you very much for your suggestion. We have added a additional classification reference for soil types.

Q4: Discussion section should be shortened and overwritten, with only relative facts kept. Much of the text relates to the general conclusions, without specific direct effects in particular experimental case. More comparisons with similar research should be made in this section. No direct synergistic mechanism of biochar and PGPR in this research was revealed. More clarification and direct connection of soil amendments and measured effects should be provided.

A4: Thank you very much for your suggestion. We have revised the discussion section to make it more comprehensive.

Round 2

Reviewer 2 Report

Manuscript is suficciently revised and should be accepted in present form.